# Left Ventricular Apical Aneurysm in a Dog—A Case Report

**DOI:** 10.3390/ani14233412

**Published:** 2024-11-26

**Authors:** Ozana Maria Hritcu, Radu Andrei Baisan, Aurelian Sorin Pasca, Florentina Daraban Bocaneti

**Affiliations:** Faculty of Veterinary Medicine, Iasi University of Life Sciences Ion Ionescu de la Brad, 700489 Iasi, Romania; ozy_dulman@yahoo.ro (O.M.H.);

**Keywords:** aneurysm, dog, heart, apical

## Abstract

This report presents a case of left ventricular apical aneurysm diagnosed postmortem in a 16-year-old mixed-breed male dog. Through histopathological evaluation, microscopic structural changes of the cardiac wall were revealed, whereas immunohistochemical analysis revealed remodeling of the extracellular matrix, with the expression of desmin, CD-31, α-sma, vimentin, MMP-2, MMP-9 and TIMP-1. Left ventricular apical aneurysm is a rare condition in dogs, characterized by myocardial remodeling and decreased contractility and strength of the cardiac wall. The histopathological changes and immunohistochemical findings suggest a predominance of necrotic lesions and less of a regenerative type.

## 1. Introduction

A cardiac aneurysm is defined as a localized dilatation of the cardiac wall, involving all three structures: endocardium, myocardium and epicardium [1]. The apical area of the left ventricle is considered to be more predisposed to such lesions since it has three layers and a vortex of muscles [2]. Left ventricular aneurysm can be either congenital or acquired [1]. The causes of acquired aneurysms often include myocardial infarction, leading to the bulging of necrotized and fibrotic areas. Several other causes have been reported, such as chronic ischemic lesions associated with cardiomyopathy, viral or idiopathic myocarditis, coronary artery disease or diseases of the connective tissue [1,3]. A differential diagnosis must always be performed between aneurysms, where the endocardium, myocardium and epicardium are thinned, with no or very few cardiomyocytes, manifesting dyskinesia or akinesia in the myocardial layer and diverticula, which represent an outpouching that includes all three layers and retains normal contractility with functional cardiomyocytes [3].

The remodeling processes that occur in aneurysms leading to outpouching, fibrosis and lack of normal contractility can be revealed by investigating the immunoexpression of various stromal or parenchymal elements, such as matrix metalloproteinases or intermediate filaments [4,5].

Matrix metalloproteinases (MMPs) are zinc (Zn)-dependent endopeptidases that degrade and remodel the extracellular matrix (ECM), both in physiological and pathological processes. Their activity is regulated by tissue inhibitors of matrix metalloproteinases (TIMPs), a family of proteins that inhibit their action in tissues by blocking substrate access [6].

Based on their substrate specificity and cellular localization, MMPs are classified as collagenases (MMP-1, -4, -8, -13), stromelysins (MMP -3, -10, and -11), gelatinases (MMP-2, -9), membrane-type MMPs (MMP-14) and others [7]. Normal myocardium possesses numerous ECM proteins, such as collagens, elastins, laminin and fibronectin. MMPs that cleave collagen include MMP-1, -2, -8, -9 and -14. MMP-2 is known to be involved in the regulation of various physiological and pathological processes and is expresses by all cardiac cells [8]. Moreover, the cardiomyocytes express MMP-2 within several organelles, such as the sarcomere, cytoskeleton, nuclei, caveolae, mitochondria and mitochondrial-associated membrane [4,8].

MMP-9 plays an important role in the cardiac tissue of humans and rodents. Specifically, its expression has been confirmed in various cell types, such as cardiomyocytes, neutrophils, macrophages, endothelial cells and fibroblasts. Moreover, MMP-9 was reported to be overexpressed during hypertension, atherosclerosis or infarction [6,9]. 

Tissue inhibitor of matrix metalloproteinases 1 (TIMP-1) is a glycoprotein expressed by a large number of cell types and is one of the molecules that actively binds to the active site of pro-MMPs, preventing their activation [10].

The cardiac tissue expresses cytoskeletal intermediate filament proteins, including type III proteins such as desmin and vimentin. Desmin is a muscle-specific intermediate filament expressed in abundance in cardiomyocytes. Its main function is to form a scaffold that extends across the entire diameter of the cardiomyocyte, linking the contractile system and the organelles, resulting in sarcolemma stabilization to the costameres [5]. 

Vimentin is mainly expressed by the fibroblasts responsible for myocardium stroma formation; however, its presence has been confirmed in pathological myofibroblasts, endothelial cells and smooth muscle cells [5]. 

CD-31 or PECAM-1 (platelet endothelial cell adhesion molecule) is a transmembrane glycoprotein, a member of the immunoglobulin superfamily, with multiple functions, such as leukocyte migration, platelet activation, vascular and endothelial responses, angiogenesis, thrombosis and the regulation of endothelial and hematopoietic cell function. It is expressed by platelets, monocytes, neutrophils, megakaryocytes, plasma cells, macrophages and lymphocytes. It is strongly expressed in the endocardial endothelium in humans, murine models and dogs [11,12,13]. 

α-sma (alpha smooth muscle actin) is expressed by cardiomyocytes during developing stages or in the early stages of cardiac remodeling following injury, being replaced afterwards by α-SKA (skeletal muscle actin) and α-CA (cardiac actin) [14]. 

Information regarding cardiac apical aneurysms in dogs is scarce, and only recently has a study reported an acquired biventricular aneurysm [15].

The aim of this report was to present a case of left ventricular apical aneurysm in a dog, incidentally identified during necropsy, along with a description of the histological changes and the immunohistochemical expression of extracellular matrix components involved in its remodeling. 

## 2. Materials and Methods

The cadaver of a 16-year-old mixed-breed male dog was referred for cremation to the Pathology Department of the Faculty of Veterinary Medicine of IULS, Romania, after having been euthanized in a private practice at the owner’s request following acute cardio-respiratory failure. The owner gave informed consent for necropsy and histopathological examination for research purposes. Unfortunately, no medical history of the patient could be provided by the owner or by the private practice that initially received the case.

A necropsy was performed, during which the following were noted: generalized stasis, pulmonary edema and a bulging fibrous sac in the apical area of the left ventricle. Tissue samples from the heart, liver, spleen, lungs and kidneys were collected for histopathological examination, and formalin-fixed paraffin-embedded (FFPE) 3-micrometer sections were stained using the Masson’s trichrome staining method. From a histopathological point of view, the main cardiac lesion consisted of a left ventricular apical aneurysm.

Next, formalin-fixed paraffin-embedded (FFPE) 3-micrometer sections of the aneurysm area underwent an immunohistochemical protocol (streptavidin biotin—peroxidase method—Novolink Polymer Detection System; Leica Biosystems, NewCastle, UK) as described by the authors in a previous study [16], with slight adaptations as follows: two deparaffination baths in xylene (20 min each), treatment with 97% ethylic alcohol for 10 min, a denatured ethylic alcohol bath for 10 min, treatment for the inactivation of peroxidase in an H_2_O_2_—methanol solution (4 parts H_2_O_2_ 3% solution and 1 part methanol) for 20 min and hydration with three successive alcohol baths of 90°, 80° and 50°, followed by distilled water for 5 min each. The proteolytic treatment was performed with a pH 6.00 citrate buffer. Following this treatment, two phosphate-buffered saline (PBS) washes (pH 7.4) were performed (5 min each), followed by a serum blocking step for 20 min (Protein serum block—Novolink Polymer Detection System; Leica Biosystems, NewCastle, UK). The primary antibodies anti-CD31 (Thermo Scientific, JC clone, 70A, 1:50, Waltham, MA, USA), anti-α sma (Novocastra, ASM-1 clone, 1:250, Newcastle upon Tyne, UK), anti-desmin (Dako, D33 clone, 1:50, Hamburg, Germany), anti-vimentin (Thermo Scientific, SP20 clone, 1:300, Waltham, MA, USA), anti-MMP-9 (1:200, sc-374029, Santa Cruz, Dallas, TX, USA) and anti-MMP-2 (1:200, Invitrogen, Rockford, IL, USA) were applied overnight at 4 °C. Treatment with diaminobenzidine was used to visualize the specific immunoreactivity. The immunolabelling procedure included negative control sections, where the primary antibodies were omitted and replaced with PBS instead. CD-31, α-sma, desmin, vimentin, MMP-2 and MMP-9 immunoreactivity were scored as previously described [16]: negative, + weak, ++ moderate and +++ strong positivity. The immunoreactivity was scored by two observers (OMH, ASP) under blind conditions. The slides were analyzed using a Leica DM750 optical microscope with an embedded camera (Leica ICC50HD) and Leica Application Suite V4 software.

## 3. Results

The necropsy (confirmed and completed using histopathology) showed generalized stasis and hemosiderosis (in the liver, kidneys, spleen, lungs and coronary arteries), pulmonary edema, splenic lipofuscinosis, hepatic and renal steatosis and interstitial lymphohistiocytic nephrosis. Regarding the heart, the gross pathology revealed a bulging fibrous sac consisting of and apical cardiac aneurysm, involving the epicardium, myocardium and endocardium. The shape of the lesion was that of a sac measuring about 0.7 cm/0.3 cm (Figure 1A). In the saccular area, all three layers of the cardiac wall were thinned and fibrous, revealing a membrane that extended outwards when pressure was applied to the left ventricle and retracted inwards when the pressure was removed (Figure 1B–D). 

Microscopically, the myocardium was thinned around the saccular lesion (Figure 2A,B), whereas some of the cardiomyocytes presented a homogenous internal structure, with no visible transversal striations. These cardiomyocytes were hyalinized, wavy or fragmented, and interspersed with thick collagen fibers. The blood vessels were either compressed or tortuous and were characterized by thickened walls. The terminal branch of the left coronary artery showed mineralization of the vascular wall, whereas the surrounding myocardium was necrotized, with dystrophic calcification and inflammatory infiltrate (Figure 2C). An increased number of small capillaries was observed in the damaged area (Figure 2D). In the central part of the aneurysm, hemosiderin deposits, mineralization areas and lipid accumulation within the fibroblasts were noted. 

Immunohistochemistry showed strong intracytoplasmic positivity for desmin in the cardiac muscle fibers (Figure 3a), positivity for α-sma in the smooth muscle fibers of the vascular walls (Figure 3b), weak positivity for CD-31 in the cardiomyocytes and positivity for vimentin in the cardiac stroma (Figure 3c). In the lateral regions of the aneurysm, irregular expression of desmin in the cardiac cells was noted, associated with clear disorganization of the transverse striations and intercalated discs. Some cells displaying an alternation of normal, wavy and fractured striations were desmin immunopositive, indicating cardiac remodeling. The desmin response was much weaker in the saccular area, where the cardiac fibers had been almost completely replaced by fibrous tissue, but vimentin expression was stronger in the fibrotic areas, both in the actual aneurysm and in the lateral regions of it, where the cardiac muscle had also undergone fibrosis.

A strong positive immunoexpression was observed in this case inside the walls of the blood vessels, indicating the presence of newly formed capillaries. Additionally, a strong immunosignal for α-sma was noted in the fibroblasts in the saccular area and in a few cardiomyocytes in the marginal regions of the aneurysm. The CD-31 response was positive, especially in the cells lining the aneurysm and in those from the new blood vessels (Figure 3d).

MMP-2 immunoexpression was strong both in the stroma and the cardiac muscle fibers, suggesting an ongoing reaction of the ECM following injury and ischemia (Figure 4a). A strong immunosignal was observed for MMP-9 in the cytoplasm of the cardiomyocytes and fibroblasts, which may suggest a possible late-stage cardiac remodeling process, where collagen synthesis is reduced (Figure 4b), whereas TIMP-1 showed positive immunoexpression only in the Eberth’s lines, which could indicate a tendency to maintain the unity between the muscle cells in order to allow for normal contractility of the myocardium (Figure 4c). No reactivity was noted in the negative control (Figure 4d).

## 4. Discussion

A cardiac aneurysm is described as a localized dilatation of the cardiac wall (congenital or acquired) involving the endocardium, myocardium and epicardium, with the absence or a low number of cardiomyocytes displaying dyskinesis or akinesis. The main cause of acquired aneurysms is usually a myocardial infarction, followed by the thinning of the necrotized, fibrotic area and bulging due to blood pressure [1].

In this case, the aneurysm was found in the apical region of the left ventricle, a location that is uncommon in dogs. Moreover, a similar lesion with the same location was reported in a cat [3], but in dogs, cardiac aneurysms have been reported in the atria [17,18], Valsalva sinus [19,20], interventricular septum [21] and interatrial septum [22]. For the case presented in this report, the necropsy revealed a left ventricular apical cardiac aneurysm, which, to our knowledge, is very rare. Only recently was a case reported in a dog, where macroscopic and microscopic findings of an aneurysm affecting both ventricles were described [15].

In humans, aneurysms are reported as post-infarction complications that are often asymptomatic and difficult to diagnose without cardiac ultrasound [23].

Studies have shown that cardiomyocytes are very sensitive to the lack of oxygen, and the ischemia associated with injury (infarction, reperfusion injury and viral myocarditis) may be responsible for necrosis and gradual replacement with fibrous tissue, leading to cardiac remodeling and reduced function [4,7,24].

Since, the patient had no previous medical history, the authors assumed that the lesion was acquired, not congenital (partially based on the age of the animal), possibly following multiple and repeated damage to the cardiac muscle, resulting in necrosis, dystrophic mineralization, subsequent replacement with fibrous connective tissue and outward bulging. Histopathology showed hyalinization of cardiac muscle fibers, mineralization and fibrosis in the lateral areas of the aneurysm and multiple fibrous pockets that began to form in the upper areas of the ventricular wall, between the cardiomyocytes. Similar findings were reported in a cardiac apical aneurysm in a cat [3].

Immunohistochemistry was performed to investigate the remodeling occurring in the aneurysm area and the cardiac wall around it.

Desmin is an intermediate filament that stabilizes the cardiomyocytes through scaffolds [5]. Immunohistochemistry showed a lack of desmin expression in the saccular region, similar to what Janus and co-workers found in 2015, suggesting a depletion of cardiomyocytes in this area [25], concurrent with the diagnosis of an aneurysm rather than a diverticulum [3].

α-sma is expressed in the early stages of cardiac remodeling following injury [14]. Some authors have indicated that the presence of α-sma in stromal fibroblasts suggests a tendency towards healing and formation of myofibroblasts [14,26]. Very few cardiomyocytes showed a weak positive response for α-sma in the sarcomeres, and authors who found similar results correlate it with the slight hypertrophy that was also observed in this case [27].

Vimentin is expressed both in normal fibroblasts that form the stroma of the myocardium and in pathological myofibroblasts [5]. In this case, vimentin expression was strongly detected in the fibrotic areas, similar to what other authors have found and correlated with the transition of the normal myocardium structure into fibroblastic and endothelial proliferation [28]. The same authors also suggested that positivity for vimentin usually indicates the presence of remodeling processes following infarction, ischemia and necrosis [25,28].

CD-31 is strongly expressed in the endocardial endothelium [12]. In this case, immunoexpression was stronger near the area affected by fibrosis, where vimentin expression was also elevated. Another study found a similar result, suggesting a tendency towards neoangiogenesis and stromal remodeling [29].

MMPs contribute to angiogenesis through various mechanisms, including the migration of endothelial cells through surrounding tissues by disrupting the ECM [30]. MMP-2 plays an important role in mediating ECM changes following cardiac injury and contributes to the acute, reversible mechanical dysfunctions that occur after reperfusion [4]. Inside the sarcomeres, MMP-2 targets troponin I, a regulator of actin–myosin activity, causing contractility dysfunctions [4,8]. Increased MMP-2 expression in cardiomyocytes, similar to the one reported in this case, has been correlated with degenerative disease in human cardiac valves [31]. Additionally, increased immunosignal was found in the sclerotic area in post-infarction myocardium in humans [32].

MMP-9 is an early and major MMP brought into the infarcted region by activated leukocytes (neutrophils and macrophages) within minutes after the ischemic lesion and is later upregulated in myocardial fibroblasts in oxidative stress [6].

Other authors have observed similar results in murine models to those reported in this case, such as increased MMP-9 expression levels in infarcted regions in cardiomyocytes and fibroblasts, and correlated them with high levels of hypoxia and ischemia, or with decreased collagen synthesis and a collagenolytic environment [6,9,33]. Hypoxia following myocardial injury increases the expression of MMP-9, but not necessarily MMP-2, which results in collagen accumulation [34]. In the case of the cardiac aneurysm presented in this study, the most probable cause was cardiac injury, such as infarction, and the hypoxia that followed could be responsible for the strong expression of both MMP-9 and MMP-2, as well as the remodeling of the tissue.

In the case analyzed here, TIMP-1 immunoexpression was weak and mostly present in Eberth’s lines between the cardiomyocytes, which could indicate a tendency towards maintaining the unity between the muscle cells to allow normal contractility of the myocardium. In humans, TIMP-1 is mainly expressed in the myocardium by fibroblasts and cardiomyocytes, and its deficiency is associated with cardiac remodeling in the left ventricle following myocardial infarction [6]. Its expression in the peri-infarction area is associated with a decreased infarcted area size and fibrosis [10], and the decrease or lack of its expression is correlated with a larger infarcted area and lower myocardial remodeling, as Creemes found in 2003 [35].

Myocardial remodeling represents a predictive factor for dilated cardiomyopathy. In the normal heart, the MMP–TIMP collagen control fluctuates, and left ventricular dilatation is thought to result from the increased collagenolytic activity of MMP–TIMP [36]. In this case, the lack of TIMP-1 expression may have contributed to the formation of the bulging aneurysm. Other authors have found that the expression of TIMP-1 in the myocardium in humans and murine models stimulates regenerative fibrosis, even independent of its interaction with matrix metalloproteinases [37].

## 5. Conclusions

This paper describes the pathological findings of a dog diagnosed with a left ventricular apical aneurysm during necropsy. Ventricular aneurysm is a rare condition in dogs, characterized by myocardial remodeling and decreased regional contractility, most likely as a consequence of myocardial infarction. Histopathological examination revealed a thin myocardium around the saccular lesion with vascular abnormalities. The necrotic lesions of the myocardium were suggested by the lack of cardiomyocytes on histopathology, the lack of desmin expression in the saccular region and the strong immunoexpression of vimentin. The fibrosis in the outpouching area of the aneurysm was not extensive and mainly affected the middle layer, where the newly formed capillaries expressing α-sma and the strong CD-31 expression in the local endocardium suggest a tendency towards reperfusion. The strong immunoexpression of MMP-2 and MMP-9, along with minimal TIMP-1 expression in the peripheral areas of the lesion, suggest ongoing remodeling changes in the myocardium.

## Figures and Tables

**Figure 1 animals-14-03412-f001:**
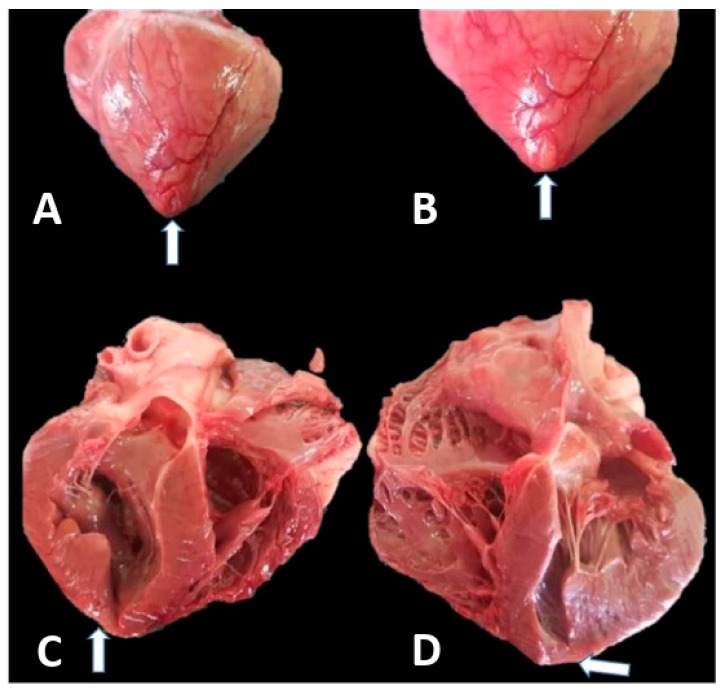
Gross pathology of the cardiac apical aneurysm. (**A**) Saccular region of the apical aneurysm without any pressure exerted on it (white arrow). (**B**) Saccular region of the apical aneurysm with pressure exerted on it, causing it to bulge out (white arrow). (**C**,**D**) Apical aneurysm transversally sectioned. The saccular region is so thinned out that it is barely visible (white arrows).

**Figure 2 animals-14-03412-f002:**
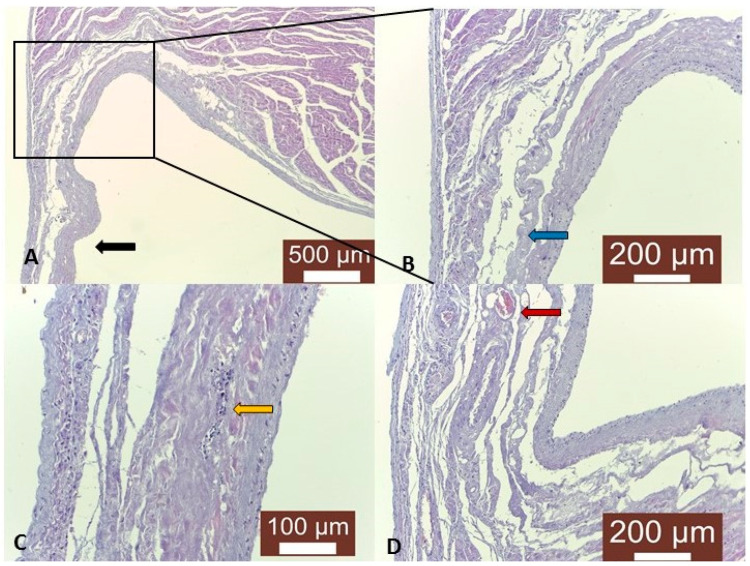
Cardiac apical aneurysm in a dog. (**A**) Thinned apical wall with fibrosis and no cardiomyocytes (black arrow), Masson’s trichrome stain, 40X; (**B**) Inset—epicardium, endocardium and thinned myocardium in the saccular region of the aneurysm, with visible fibrosis and a low number of cardiomyocytes (blue arrow), Masson’s trichrome stain, 100X. (**C**) Detail of the saccular region of the aneurysm. Fibrosis and dystrophic calcification (yellow arrow); Masson’s trichrome stain, 400X. (**D**) Multiple capillaries formed in the saccular region of the aneurysm (red arrow) and fibrosis, Masson’s trichrome stain, 100X.

**Figure 3 animals-14-03412-f003:**
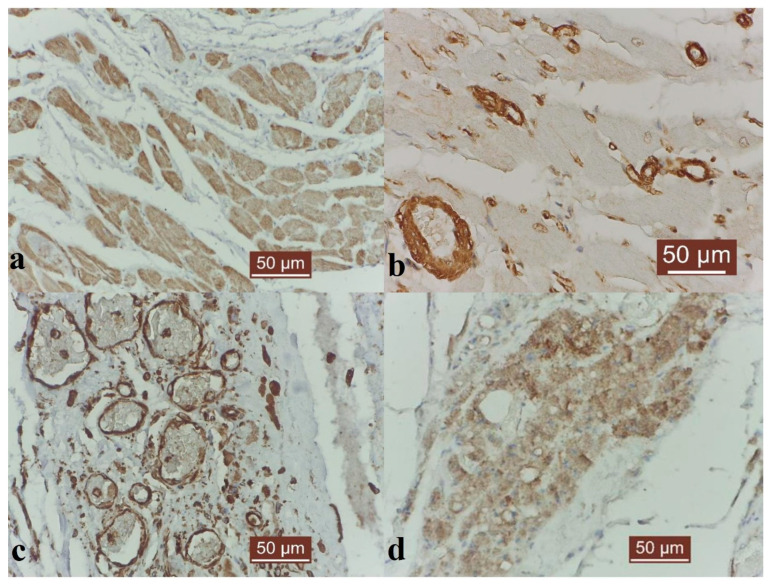
Desmin, α-sma, vimentin and CD-31 immunostaining in a dog aneurysm. (**a**) Strong immunosignal for desmin in cardiomyocytes; Hematoxylin background, 400X. (**b**) Strong immunosignal for α-sma in the smooth muscle fibers in the myocardial capillaries; Hematoxylin background, 400X. (**c**) Strong vimentin immunosignal in the endothelial cells of myocardial capillaries; Hematoxylin background, 400X. (**d**) Weak CD-31 immunosignal in cardiomyocytes; Hematoxylin background, 400X.

**Figure 4 animals-14-03412-f004:**
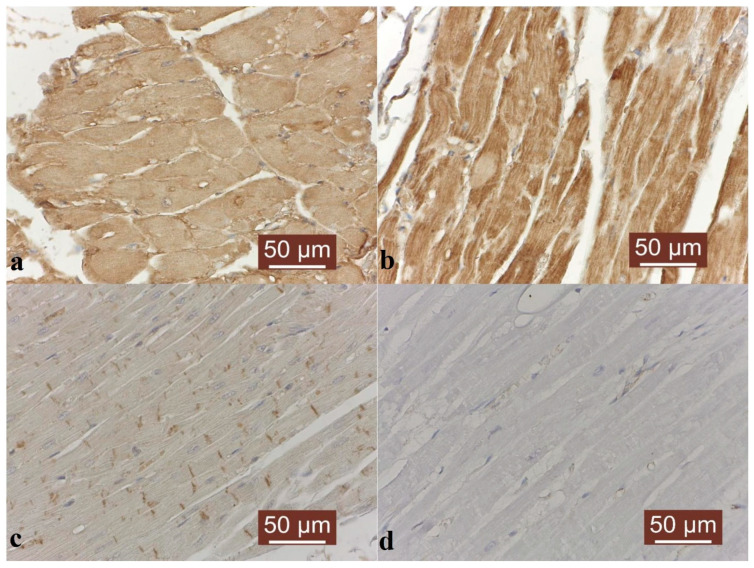
MMP-2, MMP-9 and TIMP-1 immunostaining in a dog aneurysm. (**a**) Strong immunosignal for MMP-2 in cardiomyocytes; Hematoxylin background, 400X. (**b**) Strong immunosignal for MMP-9 in cardiomyocytes; Hematoxylin background, 400X. (**c**) Weak immunosignal for TIMP-1 in cardiomyocytes. Strong immunosignal in the Eberth’s lines; Hematoxylin background, 400X. (**d**) Negative control; Hematoxylin background, 400X.

## Data Availability

The original contributions presented in this study are included in the article. Further inquiries can be directed to the corresponding author.

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
