# Peer review of "Left Ventricular Apical Aneurysm in a Dog—A Case Report"

_animals, 2024, doi:10.3390/ani14233412_

Round 1

Reviewer 1 Report

Comments and Suggestions for Authors

Dear Authors, the case report is original in its rarity but needs major revision  in many chapters of the manuscript.

Chapter 2. Materials and Methods

1) In primis is really important to justify the euthanasia. In the manuscript the case report is presented as acute cardio-respiratory failure without any clinical and strumental cardiological investigation.  Please, improve the cardiological with biochemical analysis and strumental investigations (cardio-ultrasound) data for supporting the necessity of euthanasia. It is a real important ethical aspect !

2) Please improve the description of necropsy and related ancillary investivationg (histopathology and immunohistochemistry) using appropriate lexicon in use in pathology (e.g. A complete necropsy is pleonastic it is better use Necropsy, replace harvested with collected, explain better the preparation of slide using terms as formalin-fixed-paraffin-embedded (FFPE) 3 micron sections, ect).

Please improve the IHC negative controls indicating which block system has been used for "Endogenous tissue backgraund control".

Plese, indicate the imaging capture system operating with the optical microscope.

Chapter 3 Results

IHC comments use in chapter Discussion are more appropriate if presented in Chapter - Results.

Fig. 3 captation  a - move the optical magnification after the number as follow: 40X; captation c - lacks the optical magnification after the number.

Chapter 4. - Discussion

Rewrite the chapter after repositioning pertinent  IHC comments in  Chapter 3. Results.

Chapter 5. - Conclusion

Please improve the Conclusion with more details related to cardiomyopathy (primary or  secondary; eccentric and so on).

Comments on the Quality of English Language

Improve lexicon use in pathology.

Author Response

Response to reviewer 1

We thank the referee for his/her valuable suggestions that will certainly contribute to improving the quality of our paper.

Please find below the responses to the referee, point by point.

  1. In primis is really important to justify the euthanasia. In the manuscript the case report is presented as acute cardio-respiratory failure without any clinical and strumental cardiological investigation. Please, improve the cardiological with biochemical analysis and strumental investigations (cardio-ultrasound) data for supporting the necessity of euthanasia. It is a real important ethical aspect !

Response: We are a referral veterinary hospital and only received the cadaver of the animal for cremation. The owner brought the cadaver and agreed to allow us to perform a necropsy for research purposes. During the necropsy we found the lesions that are described in the article. We contacted the private practice that performed the euthanasia of the animal, but they could only provide us with the protocol they used and the reason that was the owner’s request and poor clinical state of the animal. The owner presented the patient in severe cardio-respiratory distress and refused any investigations (cardiologic consultation, laboratory investigations) for financial reasons, and could not provide any medical history of the patient. The euthanasia was performed at the request of the owner, after being informed that that animal would have to be treated for whatever disease it may suffer from, again for financial reasons.

We cannot assume responsibility for the reason of the euthanasia since it was performed at the owner’s request and not in our clinic.

Please see lines 94-100.

  1. Please improve the description of necropsy and related ancillary investivationg (histopathology and immunohistochemistry) using appropriate lexicon in use in pathology (e.g. A complete necropsy is pleonastic it is better use Necropsy, replace harvested with collected, explain better the preparation of slide using terms as formalin-fixed-paraffin-embedded (FFPE) 3 micron sections, ect).

Response: We thank the referee for these observations and we agree that it would greatly improve the understanding of the presented case if ancillary investigations could be procured. Unfortunately, these were not performed at the owner’s request, for financial reasons. 

Please see lines 101, 104.

  1. Please improve the IHC negative controls indicating which block system has been used for "Endogenous tissue backgraund control".

Response: We have made the requested changes in the text. Please see lines 117-118.

  1. 5Plese, indicate the imaging capture system operating with the optical microscope.

We have mentioned the image capture system operating with the optical microscope. Please see line 130-131.

  1. 3 captation a - move the optical magnification after the number as follow: 40X; captation c - lacks the optical magnification after the number.

Response: We apologize for this omission. We have corrected the legend of the images. Please see lines 183-188.

  1. Chapter 3 Results

IHC comments use in chapter Discussion are more appropriate if presented in Chapter - Results.

Chapter 4. - Discussion

Rewrite the chapter after repositioning pertinent  IHC comments in  Chapter 3. Results.

Chapter 5. - Conclusion

Please improve the Conclusion with more details related to cardiomyopathy (primary or  secondary; eccentric and so on).

Response: We’ve moved the IHC comments from Discussions to the Results section. We thank the referee for this comment, indeed is provides more clarity to both sections. Please see lines 165-180, 189-196.

We have re-written the Discussion and Conclusion chapter. Please see the lines 233-242, 255-266, 293-305.

Reviewer 2 Report

Comments and Suggestions for Authors

Dear author, this case report can be interesting for a pathologist. In general I would improve the introduction and I would change completely the conclusion of this case report. The aim of your study was the histologic and immunohistological description of the aneurismatic lesion, and was not to give suggestion to veterinary cardiologist about diagnosis of cardiomyopathies. 

Line 33: “Left ventricular aneurysm can be either congenital or acquired”. Put a reference please. 

Line 35: not only hypertrophic cardiomyopathy but cardiomyopathy in general and you not reported coronary artery disease. 

Line 37: Describe better the difference between aneurism and diverticula

Line 39: I would say a “normal contractility”

Line 41: is not clear why you start to describe the matrix metalloproteinases and the other molecules. It seems not well connected to the rest of the introduction. 

Line 90: I would not say  “euthanasia is recommended”. It seems that the clinicians proposed euthanasia without having carried out any diagnostic tests to understand the reason of the cardio-respiratory failure in the dog. 

Line 126: Did you find any other cardiac or systemic abnormalities. The rest of the heart? The rest of the organs? Lung, liver, kidney? What did you find on necropsy?

Line 269: I would suggest to completely change the conclusion. The aim of the study was not to give suggestion to veterinary cardiologist about the diagnosis of cardiomiopathies, but to describe the histology and immunohistochemical expression of the extracellular matrix. You do not evaluate cardiac contractility (both directly or indirectly) you just describe the macroscopic and microscopic lesion of a dog with a left ventricular apical aneurism. 
Therefore, I would not make any conclusions regarding the clinical and diagnostic approach to be used in patients with cardiomyopathies but would limit  to describe the histological and immunohistochemical findings and made conclusion base on them. This case completely lacks clinical and diagnostic description. 

Generally aneurism are not a differential diagnosis of cardiomyopathy but are a consequences of a cardiomyopathy or coronary disease.

The conclusion should be change also in the abstract session. 

Author Response

Response to reviewer 2

We thank the referee for his/her valuable suggestions that will certainly contribute to improving the quality of our paper.

Please find below the responses to the referee, point by point.

  1. Line 33: “Left ventricular aneurysm can be either congenital or acquired”. Put a reference please.

Response: We thank the referee for this observation. We have added the required reference. Please see line 34.

  1. Line 35: not only hypertrophic cardiomyopathy but cardiomyopathy in general and you not reported coronary artery disease.

Response: We appreciate the correction provided by the referee and have made the appropriate addition in the text. Please see lines 36-37.

  1. Line 37: Describe better the difference between aneurism and diverticula

Response: The suggestion made by the referee is correct and we thank for it. We have completed the differential description between aneurysm and diverticula. Please see lines 37-42.

  1. Line 39: I would say a “normal contractility”

Response: We thank the referee for this correction and we have also performed the correction in the text. Please see line 41.

  1. Line 41: is not clear why you start to describe the matrix metalloproteinases and the other molecules. It seems not well connected to the rest of the introduction.

Response: The referee is right in its observation and we apologize for this omission.  We have added an explanation in the text that should function as a continuation between the two introductory parts. Please see lines 43-46.

  1. Line 90: I would not say “euthanasia is recommended”. It seems that the clinicians proposed euthanasia without having carried out any diagnostic tests to understand the reason of the cardio-respiratory failure in the dog.

Response: We thank the referee for this observation. We are a referral veterinary hospital and only received the cadaver of the animal for cremation. The owner brought the cadaver and agreed to allow us to perform a necropsy for research purposes. During the necropsy we found the lesions that are described in the article. We contacted the private practice that performed the euthanasia of the animal, but they could only provide us with the protocol they used and the reason that was the owner’s request and poor clinical state of the animal. The owner presented the patient in severe cardio-respiratory distress and refused any investigations (cardiologic consultation, laboratory investigations) for financial reasons, and could not provide any medical history of the patient. The euthanasia was performed at the request of the owner, after being informed that that animal would have to be treated for whatever disease it may suffer from, again for financial reasons.

We cannot assume responsibility for the reason of the euthanasia since it was performed at the owner’s request and not in our clinic.

Please see lines 94-100.

  1. Line 126: Did you find any other cardiac or systemic abnormalities. The rest of the heart? The rest of the organs? Lung, liver, kidney? What did you find on necropsy?

Response: The referee is right in his/her request. We apologize for omitting the other findings of the necropsy. However, the necropsy did not reveal much information and that is why we chose to omit it. We found generalized stasis and hemosiderosis, acute pulmonary edema, liver and renal steatosis, interstitial lymphohistiocytic nephrosis and spleen lipofuscinosis. We have added this information to the description in the article. Please see lines 101-102, 133-136.

  1. Line 269: I would suggest to completely change the conclusion. The aim of the study was not to give suggestion to veterinary cardiologist about the diagnosis of cardiomiopathies, but to describe the histology and immunohistochemical expression of the extracellular matrix. You do not evaluate cardiac contractility (both directly or indirectly) you just describe the macroscopic and microscopic lesion of a dog with a left ventricular apical aneurism.

Therefore, I would not make any conclusions regarding the clinical and diagnostic approach to be used in patients with cardiomyopathies but would limit  to describe the histological and immunohistochemical findings and made conclusion base on them. This case completely lacks clinical and diagnostic description. Generally aneurism are not a differential diagnosis of cardiomyopathy but are a consequences of a cardiomyopathy or coronary disease.

The conclusion should be change also in the abstract session.

Response: We thank the referee for this suggestion. We have changed the conclusion. Please see lines 293-305.

Reviewer 3 Report

Comments and Suggestions for Authors

I would like to congratulate authors for this manuscript. It is very well written, concise and precise. The description of component of cardiac tissue  cytoskeleton is very detailed, and so is the immunohistochemistry procedures performed.

On the other hand, it would be interesting to know about the condition than cause the clinical signs of the patient and that lead him to be euthanized. A brief description of the dog condition antemortem, and If any tests were performed, x-rays, echocardiography, the results of those and the decision to euthanized the patient.

In the bibliography section there is a minor mistake, editing mistake, where Reference 37 (line 378) and reference 39 (line 381) are blank, but there is a Reference 38, not mention on the main text. I guess reference 37 on bibliography is Reference 38, and Reference 38 and 39 do not exist.

Line 15-16 “clinicians should be aware the ventricular aneurysm”, should it say “clinicians should include”

Line 62 “The cardiac tissue is expressing…”, would not it be better: “The cardiac tissue expresses…”?

Lines 196 and 255 “in between”, it would be better just “between”.

Author Response

Response to reviewer 3

We thank the referee for his/her valuable suggestions that will certainly contribute to improving the quality of our paper.

Please find below the responses to the referee, point by point.

  1. On the other hand, it would be interesting to know about the condition than cause the clinical signs of the patient and that lead him to be euthanized. A brief description of the dog condition antemortem, and If any tests were performed, x-rays, echocardiography, the results of those and the decision to euthanized the patient.

Response: We thank the referee for this observation. We are a referral veterinary hospital and only received the cadaver of the animal for cremation. The owner brought the cadaver and agreed to allow us to perform a necropsy for research purposes. During the necropsy we found the lesions that are described in the article. We contacted the private practice that performed the euthanasia of the animal, but they could only provide us with the protocol they used and the reason that was the owner’s request and poor clinical state of the animal. The owner presented the patient in severe cardio-respiratory distress and refused any investigations (cardiologic consultation, laboratory investigations) for financial reasons, and could not provide any medical history of the patient. The euthanasia was performed at the request of the owner, after being informed that that animal would have to be treated for whatever disease it may suffer from, again for financial reasons.

We cannot assume responsibility for the reason of the euthanasia since it was performed at the owner’s request and not in our clinic.

Please see lines 94-100.

  1. In the bibliography section there is a minor mistake, editing mistake, where Reference 37 (line 378) and reference 39 (line 381) are blank, but there is a Reference 38, not mention on the main text. I guess reference 37 on bibliography is Reference 38, and Reference 38 and 39 do not exist.

Response: We apologize for these mistakes and thank the referee for the correction. We have adressed these issues in the text. Please see lines 404-408.

  1. Line 15-16 “clinicians should be aware the ventricular aneurysm”, should it say “clinicians should include”

Response: We thank the referee for this suggestion. We have changed this sentence with a short conclusion in the manuscript. Please see lines 15-16.

  1. Line 62 “The cardiac tissue is expressing…”, would not it be better: “The cardiac tissue expresses…”?

Response: The referee is correct and we are gratefull for this suggestion. We have made the changes in the text. Please see line 68.

  1. Lines 196 and 255 “in between”, it would be better just “between”.

Response: We thank the referee for this observation. We have made the changes in the text. Please see lines 229, 277.

Reviewer 4 Report

Comments and Suggestions for Authors

The manuscript shows a very interesting case report of an apical aneurysm of the left ventricle in a dog. Unfortunately, there were no cardiological studies prior to the autopsy, which would enrich the presentation. The introduction is clear and presents the main background of the topic, which reveals the originality necessary for the presentation of a case to be valid. The materials and methods are well described; however, it would greatly enrich the comparison with a normal heart, In the results, it would be important to improve the figure shown by the trichrome staining, also including some greater magnification. The discussion is complete and orderly; It would be important for them to mention if there is a antecedent of the particular labeling with TIMP-1 in the heart of other species. An important aspect to correct is that in some parts of the text the term cardiac tissue is used; it would be important to clarify which tissue component of the heart they are referring to in each case.

Author Response

Response to reviewer 4

We thank the referee for his/her valuable suggestions that will certainly contribute to improving the quality of our paper.

Please find below the responses to the referee, point by point.

1. The materials and methods are well described; however, it would greatly enrich the comparison with a normal heart,

Response: We appreciate the observation of the referee. We have investigated the antibodies used for immunohistochemistry based on their expression in the normal heart tissue. Please see lines 54-63, 68-74, 80-85.

2.  In the results, it would be important to improve the figure shown by the trichrome staining, also including some greater magnification.

Response: We thank the referee for this suggestion. We have included a few aditional figures with higher magnification in order to enrich the histopathological description. Please see Figure 2, lines 157-164.

3. The discussion is complete and orderly; It would be important for them to mention if there is an antecedent of the particular labelling with TIMP-1 in the heart of other species.

Response: We appreciate the referee’s observation. To our best knowledge, previous immunohistochemical studies of TIMP-1 in myocardiocytes do not describe labelling in the Eberth lines in other species.

 4. An important aspect to correct is that in some parts of the text, the term cardiac tissue is used; it would be important to clarify which tissue component of the heart they are referring to in each case.

Response: We thank the referee for this observation. We have described in the text which cells are involved in the synthesis of desmin and vimentin. Please see lines 61-63, 69-70, 74.

Round 2

Reviewer 1 Report

Comments and Suggestions for Authors

Dear Authors, your manuscript is ready for publication.

Author Response

Comments

1. Dear Authors, your manuscript is ready for publication.

We thank the referee for his/her valuable suggestions and we considered them for the revised version. 

Reviewer 4 Report

Comments and Suggestions for Authors

The manuscript is in a condition to be accepted; however, two issues must be modified. On the one hand, not using the term degenerative, when in fact the lesions described show that there is necrosis and on the other hand, changing micron (a very old term) to micrometer

Author Response

The manuscript is in a condition to be accepted; however, two issues must be modified.

  1. On the one hand, not using the term degenerative, when in fact the lesions described show that there is necrosis

Response: We thank the referee for his/her valuable suggestions, and indeed all contributed to an improved version. 

We appreciate this suggestion, therefore we changed this term in the manuscript draft. Please see lines 15 and 297.

2. on the other hand, changing micron (a very old term) to micrometer

Response: We apologise for this terminology and we made the corrections, replacing it with ”micrometer”. Please see lines 107 and 107